# Marginal Utility for Planning
# in Continuous or Large Discrete Action Spaces

**Zaheen Farraz Ahmad**
Department of Computing Science
University of Alberta
zfahmad@ualberta.ca

**Levi H. S. Lelis**
Department of Computing Science
University of Alberta
levi.lelis@ualberta.ca

**Michael Bowling**
Department of Computing Science
University of Alberta
mbowling@ualberta.ca

## Abstract

Sample-based planning is a powerful family of algorithms for generating intelligent behavior from a model of the environment. Generating good candidate actions is critical to the success of sample-based planners, particularly in continuous or large action spaces. Typically, candidate action generation exhausts the action space, uses domain knowledge, or more recently, involves learning a stochastic policy to provide such search guidance. In this paper we explore explicitly learning a candidate action generator by optimizing a novel objective, marginal utility. The marginal utility of an action generator measures the increase in value of an action over previously generated actions. We validate our approach in both curling, a challenging stochastic domain with continuous state and action spaces, and a location game with a discrete but large action space. We show that a generator trained with the marginal utility objective outperforms hand-coded schemes built on substantial domain knowledge, trained stochastic policies, and other natural objectives for generating actions for sample-based planners.

## 1 Introduction

Sample-based planning is a powerful family of algorithms used in decision-making settings where the objective is to find the best action from among a number of possibilities. These algorithms involve an agent repeatedly sampling actions for evaluation in order to find an optimal action. In domains with large (possibly continuous) action spaces, the number of samples needed to identify optimal actions renders an exhaustive search intractable. In many domains, such as sequential games, the outcome of an action can be stochastic, which further inflates the search space. In this paper, we consider problem domains with continuous or very large discrete action spaces.

Monte Carlo tree search (MCTS) methods such as UCT [8] performs search in discrete action spaces by balancing between sampling promising actions to improve the estimates of their outcomes and exploring other candidate actions. MCTS algorithms, though, might fail to encounter effective actions if the action space is very large, as the algorithm is unable to consider all available actions during search [11]. Search algorithms such as MCTS can thus benefit from an action candidate generator, that given a state, returns a set of promising actions to be executed at that state.

For search in a continuous action setting, Yee et. al [18] introduced Kernel Regression UCT (KR-UCT), an MCTS algorithm for continuous spaces and stochastic outcomes that requires an action

candidate generator. Yee et. al [18] used hand-coded generators in their experiments. Meanwhile, reinforcement learning with self-play was shown to be successful at training agents to search in domains with a relatively large action space, such as Go [13, 15], using a learned policy to guide search. Lee et al. [9] then combined this idea with KR-UCT to learn an action generator for search in continuous domains by sampling a set of actions from the policy. The key insight in this work is that sampling from a good policy does not necessarily produce a good set of actions for planning.

In this paper, we introduce a novel approach that explicitly generates sets of candidate actions for sample-based planning in large action spaces. Our approach trains a neural network model while optimizing for the *marginal utilities* of the generated actions, a concept we introduce in this paper. The marginal utility measures the increase in value for generating an action over previously generated actions, thus favoring diversity amongst the generated actions. We evaluate our approach in curling, a stochastic domain with continuous state and action spaces, and in a location game with a discrete but large action space. Our results show that our generator outperforms trained policies, hand-coded agents, and other optimization objectives when used for sample-based planning.

## 2   Related Work

Monte Carlo tree search (MCTS) is a simulation-based planning method that incrementally builds a search tree of sequential states and actions. However, this method requires evaluating every action at least once, which can be infeasible in large action spaces. To address this, other variations have been introduced that try to limit the search space. One such variant is progressive widening or unpruning [5, 3], which limits the number of actions that are evaluated at a node. Additional actions are sampled only after the state has been evaluated a sufficient number of times to provide a reliable estimate. Double progressive widening [4] extends progressive widening in MCTS to settings with continuous action spaces and stochasticity. Kernel Regression with UCT (KR-UCT) [18] combines double progressive widening with information sharing to improve value estimates and consider actions outside the initial set of candidate actions. All previous methods require an initial candidate set to commence the search procedure, thus these ideas are orthogonal to the action generators we introduce in this paper.

NaïveMCTS is an MCTS algorithm that searches in domains with action spaces defined by the combination of the actions performed by "state components" [11]. Units in a real-time strategy game or the actuators of a robot are examples of state components that can be controlled by a centralized agent. NaïveMCTS samples actions to be considered in search by independently sampling component actions. The action generator schemes we introduce are more general because they can be applied to any action space, and not only to those defined by state components.

Learning methods now play a key role in sample-based planning. One example is AlphaGo Zero [15] and AlphaZero [14], which demonstrated state-of-the-art performance in Go, chess, and shogi. The AlphaZero approach foregoes most human domain knowledge for machine learning models trained from self-play games. These learned models are employed in two ways. First, Monte Carlo rollouts are replaced with learned value estimates. Second, a learned policy is used in action selection to focus samples on actions that have high probability according to the policy using a modification of PUCB [12]. Learned policies can be used as action generators by sampling actions according to the policy distribution. We show empirically that action generators trained while optimizing for marginal utilities are better suited when used with search than actions sampled by these learned policies.

## 3   Sample-Based Planning

While we are interested in investigating the sequential decision-making setting, for simplicity we constrain ourselves to planning in a single-decision problem. We define $\mathcal{S}$ to be the set of all decision states and $\mathcal{A}$ the set of all actions. There exists a deterministic mapping $\mathcal{T} : \mathcal{S} \times \mathcal{A} \rightarrow \mathcal{Z}$ such that upon taking an action $a \in \mathcal{A}$ at a state $s \in \mathcal{S}$, the agent observes an outcome $z \in \mathcal{Z}$. There is also a reward function $R : \mathcal{Z} \rightarrow \mathbb{R}$ that associates an outcome with a real-valued reward.

The $Q$-value of a state-action pair $(s, a)$ is the expected reward obtained by applying $a$ at $s$. For domains with continuous actions spaces we introduce stochasticity through an explicit *execution model* $\mathcal{E} : \mathcal{A} \times \mathcal{A} \rightarrow \mathbb{R}$. $\mathcal{E}(a, a')$ is the probability action $a'$ is executed when action $a$ is selected. This can be thought of as the agent's imprecision in executing actions due to imperfect motors or

physical skill. The $Q$-value in continuous action spaces with stochasticity is defined as,

$$Q(s, a) = \int_{a' \in \mathcal{A}} \mathcal{E}(a, a') R(\mathcal{T}(s, a')) \,. \tag{1}$$

The goal of a planner is to select actions in a given state to maximize $Q(s, a)$.

Although we restrict our investigation to a single-decision setting where an agent sees an outcome immediately after taking an action, this still can pose a challenging search problem [1], especially when $\mathcal{S}$ and $\mathcal{A}$ are continuous or discrete and very large. This simplification allows us to focus on the evaluation of candidate generators rather than rollout policies or value function approximations. Our approach can be extended to the sequential setting by recursively defining $Q$-values in terms of the values of successive state-action pairs, a direction we intend to investigate in future works. Next, we describe two sample-based planning algorithms we use in our experiments.

## 3.1 UCB

A commonly used planning method in the single-decision setting is the Upper Confidence Bounds (UCB) algorithm [2]. UCB performs efficient sampling by balancing its budget between sampling promising actions to compute reliable estimates (exploitation) and sampling newer actions to find better actions (exploration). This algorithm tracks the average rewards and the selection counts for each action and chooses an action that has the highest one-sided confidence value calculated as:

$$\arg\max_{a} \left( \bar{Q}(s, a) + C \cdot \sqrt{\frac{\log N}{n_a}} \right) \tag{2}$$

where $\bar{Q}(s, a)$ is the average sampled reward of action $a$, $n_a$ is the number of times $a$ has been sampled, $N$ is the total number of samples, and $C$ is a constant that tunes the rate of exploration. The second term ensures that actions that have not been sampled enough will still be selected.

## 3.2 KR-UCB

Kernel Regression with UCB (KR-UCB) [18] leverages information sharing to improve the estimates of the actions and add newer actions to be considered for sampling in stochastic settings. Kernel regression is a non-parametric method to estimate the value of the conditional expectation of a random variable given sampled data [10, 17]. Given a set of data points and their values, kernel regression estimates the value of a new data point as an average over all data points in the set weighted by a *kernel function*. In the stochastic domain we consider, the kernel function used is the execution model, $\mathcal{E}$, that adds uncertainty to actions. With a kernel function, K, and data set, $(a_i, r_i)_{i=1,\ldots,n}$, the approximate expected value of action $a$ is

$$\hat{Q}(s, a) = \frac{\sum_{i=0}^{n} K(a, a_i) r_i}{\sum_{i=0}^{n} K(a, a_i)} \,. \tag{3}$$

The value $\sum_{i=0}^{n} K(a, a_i)$, which we will denote as $W(a)$, is the *kernel density*. The kernel density of any action is a measure of the amount of information from all other samples available to it.

Selection of an action with KR-UCB is governed using an adaptation of the UCB formula, where the average reward of an action is approximated by its kernel regression estimate and its visit count is approximated by its kernel density. With $C$ being the exploration, the selection policy is defined as,

$$\arg\max_{a \in \mathcal{A}} \left( \hat{Q}(s, a) + C \cdot \sqrt{\frac{\log \sum_{b \in A} W(b)}{W(a)}} \right) , \tag{4}$$

Starting with an initial set of candidate actions, KR-UCB repeatedly samples actions according to Equation 4. The rewards from all outcomes observed over all samples are used to calculate an action's kernel regression estimate. Due to its information sharing scheme, as more outcomes are observed, more information is available about all actions at the state.

KR-UCB initially considers selecting only from the set of actions provided by an action generator. Once enough outcomes have been sampled from the initial set of actions, KR-UCB considers new

actions based on their likelihood of being executed when sampling another action and on the amount of new information they share with the actions already considered. As a result, KR-UCB can find better actions than it is provided as input by an action generator through these added actions. A thorough description of KR-UCB can be found in the work by Yee et. al [18].

# 4   Marginal Utility Optimization

KR-UCB was originally proposed using a hand-coded domain-specific action generator to create its initial set of candidate actions for a given state. More recently, learned policies were used to provide search guidance [14] and explicitly as an action generator for sample-based planning [9], where the set of candidate actions are sampled from the policy. These policies are typically trained to match the sampling distributions of the planner, which itself is aimed at focusing samples on actions with high value. As a result, a policy $\pi$ trained to match the planner's sampling distributions is implicitly maximizing the objective $E_{a \sim \pi(s)}[Q(s,a)]$. However, this objective ignores that the policy is being used to generate a set of candidate actions for planning. In this setting of generating a set of candidate actions it is only important that some sampled actions have high value rather than all sampled actions.

In place of a policy, we propose explicitly optimizing a candidate action generator for sample-based planning. Let $g(s) \subset \mathcal{A}$ be a generator function that given a state $s$ returns a subset of $m$ candidate actions, i.e., $|g(s)| = m$. As noted above, a policy as a generator is implicitly optimized for

$$u_{\text{SUM}}(g|s) = \frac{1}{m} \sum_{a \in g(s)} Q(s,a). \tag{5}$$

In this form, it is clear that this objective emphasizes that all generated actions should have high value. However, as the planner is itself aiming to identify the best action from the candidate actions, a more suitable objective would be,

$$u_{\text{MAX}}(g|s) = \max_{a \in g(s)} Q(s,a), \tag{6}$$

where one ensures identifying at least one high-valued action.

While the MAX objective may be more appropriate, it tends to have a poor optimization landscape for gradient-based methods as it results in sparse gradients. For intuition, consider the partial derivatives of the objective with respect to an action $a \in g(s)$. $\frac{\partial u_{\text{MAX}}}{\partial Q(s,a)} = 1$ only where $a = \arg\max_{a \in g(s)} Q(s,a)$ and 0 for all other actions. We will instead derive an alternative objective that is similar while being friendlier to gradient-based optimization. Let $g(s) = \{a_1, \ldots, a_m\}$ be the actions from the generator and $q_i = \max_{j \leq i} Q(s, a_j)$ be the maximizing action value for actions from the subset $\{a_1, \ldots, a_i\}$, so that $q_1 \leq q_2 \leq \ldots \leq q_m$. We can rewrite the MAX objective as

$$u_{\text{MAX}}(g|s) = q_m = q_1 + \sum_{i=2}^{m} (q_i - q_{i-1})$$

$$= Q(s, a_1) + \sum_{i=2}^{m} \left( Q(s, a_i) - \max_{j < i} Q(s, a_j) \right)^+.$$

The terms in the summation are marginal utilities, representing how much utility is gained by the planner considering one more action from the generator. However, maximizing the marginal utility of an action can be achieved either by increasing the value of that action or by decreasing the value of previously considered actions. In order to prevent learning an action generator that decreases the value of previously considered actions we propose our marginal utility objective,

$$u_{\text{MU}}(g|s) = Q(s, a_1) + \sum_{i=2}^{m} \left( Q(s, a_i) - \perp \left( \max_{j < i} Q(s, a_j) \right) \right)^+. \tag{7}$$

where $\perp$ denotes a stop gradient so that there is no encouragement to reduce the value of previously considered actions. The marginal utility objective can be thought of as a sum of a set of semi-gradients, where each action from the generator is optimized to improve on previously generated actions. This objective has more dense gradients than the MAX objective function because the partial derivatives of the utility with respect to each action, $\frac{\partial u_{\text{MU}}}{\partial Q(s, a_i)}$, equals 1 whenever $Q(s, a_i) > \max_{j < i} Q(s, a_j)$ and

0 otherwise. By contrast, only the action $a_i$ with largest $Q$-value has a non-zero gradient for the MAX function. Thus, while $u_{\text{MU}}$ has the same value and global optimum as $u_{\text{MAX}}$, we believe its landscape is more conducive to optimization.[1] We validate this empirically in Section 6. Another advantage of the MU objective functions is that it reduces the encouragement for all actions to produce similar outputs, as MU is maximized only if actions $a_i$ obtain higher values than actions $a_j$, for $i > j$. We also show empirically that MU encourages action diversity (see the supplementary material.)

### 4.1 Gradient Computation for Continuous Action Spaces

We now describe how we can use our marginal utility objective to optimize a generator function using gradient ascent in continuous domains with execution uncertainty. Consider $g$ as a parameterized function $g_\theta(s) = \{a_1, \ldots, a_m\}$, such that $a_i$ is differentiable with respect to $\theta$. For example, $g_\theta$ could be represented as a neural network. We cannot directly optimize $u_{\text{MU}}$ as $Q(s, a)$ is not known and can only be estimated with samples. We optimize a differentiable approximation of $u_{\text{MU}}$ where the values of $Q$ are approximated by KR, as shown by the following equation,

$$\hat{u}_{\text{MU}}(g_\theta|s) = \hat{Q}(s, a_1) + \sum_{i=2}^{m} \left( \hat{Q}(s, a_i) - \perp \left( \max_{j<i} \hat{Q}(s, a_j) \right) \right)^+ , \tag{8}$$

where

$$\hat{Q}(s, a) = \frac{\sum_{i=1}^{n} \mathcal{E}(a, a_i') r_i'}{\sum_{i=1}^{n} \mathcal{E}(a, a_i')}. \tag{9}$$

Here, $\{a_1', a_2', \cdots a_n'\}$ is the set of action outcomes from the sample-based planner after the execution model was applied to each selected action and $r_i' = R(\mathcal{T}(s, a_i'))$ is the associated reward from outcome $a_i'$. By using kernel regression to approximate our value function, and assuming a differentiable execution model for the kernel function, we have that $\hat{u}_{\text{MU}}(g_\theta|s)$ is differentiable with respect to $\theta$. So we can optimize $\theta$ with gradient ascent.

### 4.2 Gradient Computation for Discrete Action Spaces

In discrete-action domains, the generator $g_\theta(s)$ does not explicitly generate actions. Rather the generator produces a set of distinct policies over actions $\{\pi_1^\theta, \ldots, \pi_m^\theta\}$ from which it then samples actions $a_i \sim \pi_i^\theta$ where $i = 1, \ldots, m$. We derive a sampled gradient for $u_{\text{MU}}(g_\theta|s)$ using the standard log-derivative trick, although some care is needed to properly handle the stop gradients.

**Theorem 1** *For a generator $g_\theta(s)$ producing a set of policies over actions $\{\pi_1^\theta, \ldots, \pi_m^\theta\}$, let*

$$\nabla_\theta \widetilde{u_{\text{MU}}}(g_\theta|s) \equiv \nabla_\theta \log \pi_1^\theta(\tilde{a}_1|s) Q(s, \tilde{a}_1) + \sum_{i=2}^{m} \nabla_\theta \log \pi_i^\theta(\tilde{a}_i|s) \left( Q(s, \tilde{a}_i) - \max_{j<i} Q(s, \tilde{a}_j) \right)^+ , \tag{10}$$

*where $\tilde{a}_i$ is an action sampled from policy $\pi_i^\theta$. Then, $\mathbb{E}\left[ \nabla_\theta \widetilde{u_{\text{MU}}}(g_\theta|s) \right] = \nabla_\theta u_{\text{MU}}(g_\theta|s)$.*

The proof of Theorem 1 can be found in the supplementary materials.

## 5 Experimental Setup

### 5.1 Hammer Shots in Curling

Following previous work [1, 18, 9], we use curling as the test bed for evaluating the performance of marginal utility maximization for generating actions for search in the single-decision setting with continuous actions. Curling is a sport that can be modeled as a two-player, zero-sum game with continuous state and action spaces with uncertainty in the action outcomes. The game is played between two teams of players on a sheet of ice. Typically a game consists of 10 rounds called ends. In each end, teams alternate taking shots, sliding rocks from one end of the sheet to the other, targeting

a demarcated region called the house. After all shots, the score is tallied based on the number of rocks in the house closer to the center than any of the opponent's rocks. The team with the highest cumulative points after all ends is declared the winner. The last shot of the end is called the hammer shot. We restrict our investigation to hammer shots and evaluate the algorithms on the objective of finding an action that maximizes the expected number of points scored in a hammer state.

We model a hammer shot problem as a single-decision problem where $\mathcal{S}$ is the set of all possible hammer states, $\mathcal{A}$ is the set of actions available at each state, $\mathcal{Z}$ is the set of possible outcomes for playing an action at a state and $\mathcal{R}$ is a function mapping shot outcomes to scores. Actions in curling are parameterized by 3 variables: the velocity ($\nu$) of the shot, the angle ($\phi$) of trajectory, and the turn on the shot, which is a binary value in the set $\{-1, 1\}$ indicating whether the rock is rotating clock- or counter-clockwise. The turn of the rock causes it to "curl" right or left. Every time an action is taken, uncertainty in execution results in perturbations of the velocity and angle such that we play an action with velocity, $(\nu + \epsilon_\nu)$ and angle, $(\phi + \epsilon_\phi)$. In our setting, we select a constant turn parameter — we can evaluate the other turn by laterally inverting the state configurations due to symmetry. All state transitions were done by a curling simulator built using the Chipmunk simulator [1].

## 5.2  A Location Game

We present a two-player *location game* to evaluate the marginal utility objective in a setting with discrete actions. The game states are represented as an $(n \times n)$ grid of cells where each cell has an associated real value such that sum of values over all cells equals to 1. $\mathcal{S} = \mathbb{R}^{n \times n}$ and player $i$'s action is the selection of $k_i$ cells in the grid i.e., $\mathcal{A} = \{1, \ldots, n \times n\}^{k_i}$. After actions are selected, each location receives a value equal to the sum of the values of the cells closest to it in terms of their Manhattan distance and the reward of an action is the sum of the values of the constituent locations. Cells have their values equally shared among equidistant locations. $\mathcal{R}$ is a mapping from a $(state, action)$ pair to a reward in $[0, 1]$. We investigate the case of a $10 \times 10$ grid where we must select 3 locations and an opponent selects 2 locations. The opponent deterministically selects the 2 highest valued cells reducing the game to a single-agent search problem with large discrete action space ($10^6$ actions at each state). The cells have values drawn from an inverse gamma distribution, with shape and scale parameters of $\alpha = 3$ and $\beta = 1$, respectively, and normalized to sum to 1.

## 5.3  Training an Action Generator

In both settings we used neural networks for our generator function to learn mappings from states to sets of actions. In curling, the state representations passed to the network depicted the configuration of the curling rocks in the house and the local area near it. The input state consisted of a stack of three $(250 \times 125)$ matrices: two binary matrix representing the positions of the non-hammer team's and the hammer team's rocks, and a matrix depicting the region covered by the house. The input was passed through two convolutional layers. Each layer used 32 $(5 \times 5)$ filters and max pooling with a stride of 2. The output is processed by two feed-forward layers with 2,048 and 1,024 hidden units, respectively. Hidden layers used Rectified Linear Unit (ReLU) activations and the output layer used a sigmoid activation function. The output was a vector of action parameters for velocity and angle.

The neural network weights were initialized with Glorot and Bengio's method [6]. A forward pass through the neural network produces a set of actions. These actions are passed to the curling simulator and evaluated by sampling 4 outcomes of each. Using the observed outcomes and their corresponding values, the kernel regression estimates of the values of the actions are calculated (Equation 4) and their marginal utilities are computed (Equation 7). The network weights, $\theta$, are then updated by applying gradient ascent on $u_{\mathrm{MU}}$. We trained on 640,000 hammer states for 20,000 iterations (approximately 3 days of training time on a single GTX 970 GPU). We used a mini-batch size of 32 inputs, Adam optimizer [7] with a learning rate $\alpha = 10^{-4}$ and L2-regularization on the weights with $\lambda = 0.0001$.

For the location game, the input state to the network was a matrix representing the grid. The input was passed through two convolutional layers with 32 $(3 \times 3)$ filters using ReLU activations and a single feedforward layer using a sigmoid activation. A forward pass through this network produced 8 policies from which to sample 8 actions. Actions were sampled at each state from these policies and the reward of each action observed. The network was trained by gradient ascent using Equation 10. We trained on 2,400,000 states over 75,000 iterations ($\approx$20 hours training time on a single GTX 970 GPU) with mini-batch size of 32, learning rate $\alpha = 10^{-4}$ and L2-regularization with $\lambda = 0.0001$.

## 5.4 Competing Algorithms

In curling, we compare the performance of our proposed action generator, trained to maximize total marginal utility, $u_{\text{MU}}$, against models trained with other objective functions and with other action generation approaches. The first being a set of agents that use hand-tailored action generators of different 'skill levels' built with substantial curling-specific knowledge [18]. We also compare against three other objectives: MAX ($u_{\text{MAX}}$), SOFTMAX ($u_{\text{SOFTMAX}}$, a smoothed version of MAX), and SUM ($u_{\text{SUM}}$) utility functions. These models employed the same neural network architecture described above. The softmax function has a temperature parameter, which we set to $0.1$ after performing a parameter sweep. As discussed in Section 4, previous learning systems such as AlphaZero [14] implicitly maximize $u_{\text{SUM}}$. Additionally, in order to explicitly introduce *diversity* in the actions generated, we use a different objective that modifies $u_{\text{SUM}}$ to force the actions to spread. We will call this objective $u_{\text{DIVERSITY}}$ and it is defined as

$$u_{\text{DIVERSITY}} = \sum_i \left[ Q(s, a_i) - \rho \sum_{j \neq i} \mathcal{E}(a_i, a_j) \right],$$

where $\mathcal{E}(a_i, a_j)$ is the probability that action $a_j$ is executed when attempting $a_i$. This places a penalty on the utility achieved by the actions for being close to each other and so encourages generation of variegated actions. The magnitude of this penalty is scaled using $\rho$ and influences the degree to which the actions are spread — the higher the penalty, the more diverse the actions generated. In our experiments, $\rho$ was set to $0.1$ after a parameter sweep over different values.

The other approach we compare against is an agent that uses a learned selection policy model trained using deep reinforcement learning and KR-UCB in accordance to the work of Lee et.a l [9]. The policy-based model's network output a $(32 \times 32)$ policy, $\boldsymbol{\rho}$, over a discretization of the action space. The network parameters, $\theta$, are optimized using a regularized cross-entropy loss objective with L2 regularization to maximize the similarity between policy output and the sample distributions, $\boldsymbol{\pi}$: $-\boldsymbol{\pi} \log \boldsymbol{\rho} + \lambda \|\theta\|^2$. Lee et. al [9] defined hyper-parameter values for their implementation. However, since our simulator and network are different, we searched for good values for these parameters. We found good performance by setting $\lambda = 0.0001$, having the algorithm produce 32 candidate actions, and using a budget of 128 samples for obtaining the sample distribution $\boldsymbol{\pi}$.

In the location game, we compare the performance of the generator trained using the $u_{\text{MU}}$ objective, with those trained using $u_{\text{MAX}}$ and $u_{\text{SOFTMAX}}$. We also compared its performance against REINFORCE [16] which uses a network, parameterized by $\theta$, to output a single policy from which 8 actions were sampled to update the parameters. The policy trained with REINFORCE implicitly maximizes $u_{\text{SUM}}$. After a parameter sweep, the learning rate for training with these objectives were set to $\alpha = 10^{-4}$ for $u_{\text{MAX}}$ and $\alpha = 10^{-5}$ for $u_{\text{SOFTMAX}}$ and REINFORCE.

## 6  Results and Analysis

To evaluate the performance of the various methods in curling, we used each to generate a set of actions for 3,072 test states. As outcomes are stochastic in this setting, we use UCB and KR-UCB as the planners for selecting an action. The action $a$ returned by the planning procedure is then evaluated by averaging the number of points obtained by executing $a$ 1,000 times in our curling simulator. For each method we report the expected number of points scored for the 3,072 test states with 95% confidence intervals of our approximations. For the location game, we used each of the trained generators to produce actions for 510 test states and selected the generated action that achieved the highest utility. This is achieved by computing the utility for each generated action and is the same as applying a sample-based planner as there is no stochasticity in this domain. We then calculate the expected utility achieved by each generator over all states. We first discuss the results of our experiments in curling and then we follow with a discussion on our experiments in the location game. In addition, further results are provided in the supplementary material that show optimizing for marginal utility encourages diversity in the candidate actions produced by generators.

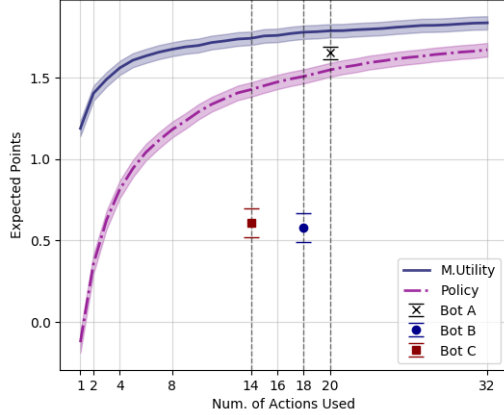

Figure 1: Expected points with UCB on actions produced by the generators in curling.

## 6.1 Results in a Continuous Setting

Figure 1 shows the expected points (y-axis) of an action generator trained by optimizing $u_{\mathrm{MU}}$, of hand-crafted action generators [18] (Bots A, B, and C), and of the generator derived from Lee et. al's policy [9] (Policy). The shaded regions and error bars denote the 95% confidence intervals. We vary the number of actions generated for planning (x-axis) from 1 to 32. The number of actions produced by the hand-crafted approaches vary from state to state, but it is fixed for a given state. We show a single point in the plot for Bots A, B, and C, where the number of actions is the average number of actions generated by the bots over all test states. Marginal utility approach significantly outperforms all methods when used with UCB. Most notably, UCB performs drastically better when using a much smaller set of generated actions from marginal utility than from the policy. The marginal utility objective results in an expected gain of over 0.15 points per hammer state over the previous policy methods. As noted by Yee et. al [18], this can lead to over a full point increase in score in a standard 10-end game of curling, and a significant increase in the likelihood of winning the game.

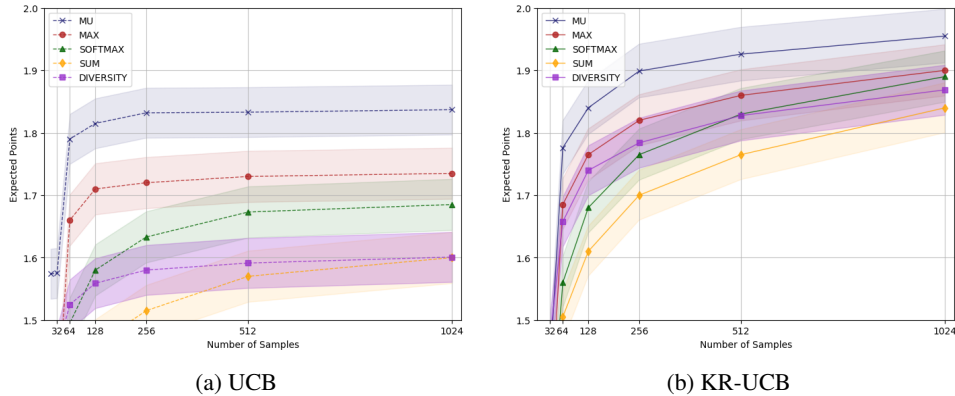

(a) UCB

(b) KR-UCB

Figure 2: Performance of search using candidate actions produced by the different models in curling.

Figures 2a and 2b plots the expected points of the generators trained on different objectives ($u_{\mathrm{MU}}$, $u_{\mathrm{MAX}}$, $u_{\mathrm{SOFTMAX}}$, $u_{\mathrm{SUM}}$, $u_{\mathrm{DIVERSITY}}$) using UCB and KR-UCB respectively. The shaded regions denote the 95% confidence intervals. Both search methods were run for a total of 1,024 sample iterations and evaluated after iterations 16, 32, 64, 128, 256, 512, and 1,024. All generators were trained to produce 32 actions. The figures show the expected number of points (y-axis) and the number of search samples (x-axis) for UCB using action generators trained while optimizing for different objective functions. Both planners initialized with actions from the marginal utility model significantly outperforms planning with other generators. In particular, these results support our hypothesis that $u_{\mathrm{MU}}$ yields an optimization landscape that is easier to optimize than $u_{\mathrm{MAX}}$. Additionally, while KR-UCB improves

upon the actions generated by all the models, marginal utility still has considerably better performance over the other generators at smaller search budgets. The actions generated by marginal utility are likely in close proximity in the action space to high-valued actions, thus allowing KR-UCT to find good actions even with a small number of samples.

## 6.2 Results in a Discrete Setting

Figure 3 plots expected utility of the actions selected using the generators trained using different objectives where higher utility signifies better performance. The lines at the top of the bars denote the 95% confidence intervals and the dotted line marks the average optimal utility over all states. In the discrete domain, the generators trained with $u_{\text{MU}}$ and $u_{\text{MAX}}$ perform equally well and the actions generated by these achieve a significantly higher utility than the actions of the other generators. This supports one of our key claims that to generate actions for planning, training a policy is not a suitable objective. Furthermore, when the optimization problem is not difficult, training either objective ($u_{\text{MU}}$ or $u_{\text{MAX}}$) can lead to good candidate generation.

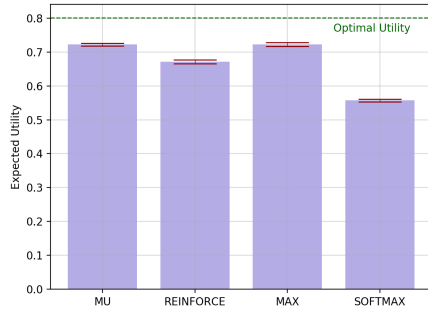

Figure 3: Expected utility of different action generators in the location game.

## 7 Conclusion

We presented a novel approach to training a candidate action generator for sample-based planning. We described how the commonly used policy-based method tries to optimize an objective that is less suited for learning actions to be candidates in planning. We proposed a method that directly tries to optimize the planner's objective of finding maximally-valued actions. We derived an intuitive reformulation of this objective, based on marginal utility, that is more conducive to optimization We showed in a location game and in the olympic sport of curling that learning systems optimizing for marginal utilities produce better search candidate actions than policy-based methods, other natural objective functions, and hand-tailored action candidates

## Impact Statement

Our work is likely to increase the overall robustness and efficiency of a broad range of fundamental search algorithms used in decision making. These search algorithms have been applied to a wide array of strategic domains such as security, scheduling and routing. Highly performant search provides improved strategic capabilities to those who employ it, lending them a competitive edge in their respective domains. As such, the societal impacts, both beneficial and harmful, of search algorithms depend on how these tools are used. We may observe increased social welfare if efficient search is used to optimize the allocation of vital resources, such as medical equipment during a pandemic. On the other hand, if accessible to those serving their own self-interests, they may monopolize markets and adversely affect the economy.

## Acknowledgement and Disclosure of Funding

This research was funded by Amii, the Alberta Machine Intelligence Institute, and compute resources were provided by Compute Canada and Calcul Québec. We would also like to thank Edward Lockhart for early discussions that led to this work.

## Footnotes

[1]Our alternative loss follows the tradition of other optimization-friendly proxy objectives, e.g., cross-entropy instead of 0-1 loss or the temporal difference sub-gradient instead of Bellman residual, which are used for their strong empirical performance.

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
