[Supplementary Material]

# A  Proof of Theorem 1

**Theorem 1** *For a generator $g_\theta(s)$ producing a set of policies over actions $\{\pi_1^\theta, \ldots, \pi_m^\theta\}$, let*

$$\nabla_\theta \widetilde{u_{\text{MU}}(g_\theta}|s) \equiv \nabla_\theta \log \pi_1^\theta(\tilde{a}_1|s)Q(s,\tilde{a}_1) + \sum_{i=2}^m \nabla_\theta \log \pi_i^\theta(\tilde{a}_i|s)\left(Q(s,\tilde{a}_i) - \max_{j<i} Q(s,\tilde{a}_j)\right)^+,$$

(1)

*where $\tilde{a}_i$ is an action sampled from policy $\pi_i^\theta$. Then, $\mathbb{E}\left[\nabla_\theta \widetilde{u_{\text{MU}}(g_\theta}|s)\right] = \nabla_\theta u_{\text{MU}}(g_\theta|s)$.*

In order to understand the sampled gradient update we need to understand how expectations interact with stop gradients ($\perp$).

**Remark 1** *We can use iterated expectation to remove the stop-gradient in a partial derivative.*

$$\begin{aligned}
\nabla_\theta \mathbb{E}[f_\theta(\perp(X_\theta))] &= \nabla_\theta \mathbb{E}[\mathbb{E}[f_\theta(\perp(X))|X = X_\theta]] \\
&= \mathbb{E}[\nabla_\theta \mathbb{E}[f_\theta(\perp(X))|X = X_\theta]] \\
&= \mathbb{E}[\nabla_\theta \mathbb{E}[f_\theta(X)|X = X_\theta]] \\
&= \mathbb{E}[\mathbb{E}[\nabla_\theta f_\theta(X)|X = X_\theta]]
\end{aligned}$$

We can now present the proof that the sampled gradient is unbiased.

**Proof of Theorem 1.**  We begin by applying Remark 1 to the unsampled gradient.

$$\begin{aligned}
\nabla_\theta u_{\text{MU}}(g_\theta|s) &= \nabla_\theta \mathbb{E}\left[Q(s,a_1) + \sum_{i=1}^m \left(Q(s,a_i) - \perp\left(\max_{j<i} Q(s,a_j)\right)\right)^+\right] \\
&= \nabla_\theta \mathbb{E}[Q(s,a_1)] + \sum_{i=2}^m \nabla_\theta \mathbb{E}\left[\left(Q(s,a_i) - \perp\left(\max_{j<i} Q(s,a_j)\right)\right)^+\right] \\
&= \nabla_\theta \mathbb{E}[Q(s,a_1)] + \sum_{i=2}^m \nabla_\theta \mathbb{E}\left[\mathbb{E}\left[\left(Q(s,a_i) - \perp\left(\max_{j<i} Q(s,a_j)\right)\right)^+\bigg|a_{i,\ldots,i-1}\right]\right] \\
&= \nabla_\theta \mathbb{E}[Q(s,a_1)] + \sum_{i=2}^m \mathbb{E}\left[\nabla_\theta \mathbb{E}\left[\left(Q(s,a_i) - \left(\max_{j<i} Q(s,a_j)\right)\right)^+\bigg|a_{i,\ldots,i-1}\right]\right]
\end{aligned}$$

Let us now look at our expected sampled gradient.

$$\mathbb{E}\left[\nabla_\theta \widetilde{u_{\text{MU}}(g_\theta}|s)\right]$$

$$\equiv \mathbb{E}\left[\nabla_\theta \log \pi_1^\theta(\tilde{a}_1|s)Q(s,\tilde{a}_1) + \sum_{i=2}^m \nabla_\theta \log \pi_i^\theta(\tilde{a}_i|s)\left(Q(s,\tilde{a}_i) - \max_{j<i} Q(s,\tilde{a}_j)\right)^+\right]$$

$$= \mathbb{E}\left[\nabla_\theta \log \pi_1^\theta(\tilde{a}_1|s)Q(s,\tilde{a}_1)\right] + \sum_{i=2}^m \mathbb{E}\left[\nabla_\theta \log \pi_i^\theta(\tilde{a}_i|s)\left(Q(s,\tilde{a}_i) - \max_{j<i} Q(s,\tilde{a}_j)\right)^+\right]$$

$$= \sum_{a_1} \pi_1^\theta(a_1|s)\nabla_\theta \log \pi_1^\theta(a_1|s)Q(s,a_1)$$

$$+ \sum_{i=2}^m \sum_{a_1,\ldots,a_i} \left(\prod_{j<i} \pi_j^\theta(a_j|s)\right) \pi_i^\theta(a_i|s)\nabla_\theta \log \pi_i^\theta(a_i|s)\left(Q(s,a_i) - \max_{j<i} Q(s,a_j)\right)^+$$

Using the log-trick, we know $\nabla_\theta \pi_i^\theta(a_i|s) = \pi_i^\theta(a_i|s) \frac{\nabla_\theta \pi_i^\theta(a_i|s)}{\pi_i^\theta(a_i|s)} = \pi_i^\theta(a_i|s)\nabla_\theta \log \pi_i^\theta(a_i|s)$. Substituting, we get,

$$= \sum_{a_1} \nabla_\theta \pi_1^\theta(a_1|s)Q(s,a_1)$$

$$+ \sum_{i=2}^{m} \sum_{a_1,\ldots,a_i} \left( \prod_{j<i} \pi_j^\theta(a_j|s) \right) \nabla_\theta \pi_i^\theta(a_i|s) \left( Q(s,a_i) - \max_{j<i} Q(s,a_j) \right)^+$$

$$= \nabla_\theta \sum_{a_1} \pi_1^\theta(a_1|s)Q(s,a_1)$$

$$+ \sum_{i=2}^{m} \sum_{a_1,\ldots,a_{i-1}} \left( \prod_{j<i} \pi_j^\theta(a_j|s) \right) \nabla_\theta \sum_{a_i} \pi_i^\theta(a_i|s) \left( Q(s,a_i) - \max_{j<i} Q(s,a_j) \right)^+$$

$$= \nabla_\theta \mathbb{E}[Q(s,a_1)]$$

$$+ \sum_{i=2}^{m} \sum_{a_1,\ldots,a_{i-1}} \left( \prod_{j<i} \pi_j^\theta(a_j|s) \right) \nabla_\theta \mathbb{E}\left[ \left( Q(s,a_i) - \max_{j<i} Q(s,a_j) \right)^+ \Big| a_{1\ldots i-1} \right]$$

$$= \nabla_\theta \mathbb{E}[Q(s,a_1)] + \sum_{i=2}^{m} \mathbb{E}\left[ \nabla_\theta \mathbb{E}\left[ \left( Q(s,a_i) - \max_{j<i} Q(s,a_j) \right)^+ \Big| a_{1\ldots i-1} \right] \right]$$

$$= \nabla_\theta u_{\text{MU}}(g_\theta|s).$$

.

## B  Diversity of Generated Continuous Actions

(a) $u_{\text{MU}}$

(b) Policy

(c) $u_{\text{MAX}}$

(d) $u_{\text{DIVERSITY}}$

Figure B.1: Coverage of actions generated by methods trained on different objectives.

To further understand the behavior of the models, we can examine the 'coverage' of the actions produced by the generators over the 2-dimensional action space (Figure B.1). We define coverage as

the region of the action space in which a particular action is generated. Each subplot illustrates the coverage of actions generated by a model. To produce these plots, we calculated the densities over the locations of each action over all test cases. The area covered by a particular action is illustrated as a shaded region where darker regions are areas in which an action is more likely to be generated.

Figure B.1a plots the coverage of the action candidates produced by the marginal utility model over all test cases. For $u_{\text{MU}}$ different actions cover separate regions in the space, i.e., the candidates generated by marginal utilities are *diverse*. This contrasts with the actions generated by the policy (Figure B.1b), which are all concentrated in one region of the action space; if one needs to find an action $a$ outside this region of the action space, it is unlikely the actions generated by the policy will be near $a$. Search algorithms such as UCB will not find $a$ as UCB is restricted to searching amongst the initial set of generated actions. KR-UCB might find and return $a$, but it can require a large number of samples to find $a$.

As observed in Figure B.1c the model trained while optimizing for $u_{\text{MAX}}$ produces a diverse set of candidate actions. However, likely due to the low density of gradients for $u_{\text{MAX}}$, the actions produced are specialized in the sense that they cover a small portion of the space. The diversity of actions generated by MAX can be helpful as they provide a diverse set of starting points for KR-UCB but could fail to generate a good set of actions for search algorithms such as UCB.

Figure B.1d illustrates the coverage of the actions generated by the objective explicitly tyring the encourage diversity. While the actions generated cover two very different regions of the action space, the actions themselves group into clusters within these regions. The amount of clustering is affected by the tuning parameter. However increasing the diversity in this setting comes at a cost to the overall performance of search.

## C  Action Ordering Learnt Using Marginal Utility

Figure C.2: Frequency of action being selected after search.

When using marginal utility as a training objective, the gradients on the actions depend on the order in which they are produced. As a result, there is an inherent ordering over the actions being produced. Figure C.2 plots the frequency at which actions have been selected during evaluation. That is, we plot the counts of the actions selected after search over all test cases. As can be seen, the action that is selected most frequently is the action that is generated first. There is potential in using this ordering in the selection policy or for progressive widening of actions in search.