[Reviews · NeurIPS 2020]

Review 1

Summary and Contributions: Post-rebuttall: I would like to thank the authors for their response. I am still in favor of accepting this paper. ============================= This paper addresses sample-based search (e.g., MCTS) in problems with a large branching factor (continuous actions or a large number of discrete actions). It proposes a new candidate action generator (a procedure which samples the applicable actions) that can be used by a higher level search algorithm. This generator is based on the observation that we want to have the best possible action in the sample, and we don't care about the quality of the other actions, thus we want to optimize the MAX quality of the sampled actions, rather than the SUM. However, optimizing for MAX is difficult, since the gradients are sparse. The main technical contribution is an elegant rewriting of the MAX objective in terms of the marginal utility of adding a new action to the sample, called MU. Optimizing the MU objective is easier, because the gradients are less sparse, and the empirical results demonstrate this well.

Strengths: The paper is well written, and very relevant to NeurIPS. I believe this paper will be of interest to a significant part of the community. The proposed candidate generator is novel, and results in good performance in the empirical evaluation.

Weaknesses: The main limitation of this paper is that it only addresses single-shot decision making (e.g., the last play in a game), where the rewards of each action are sampled directly. Putting this into a full MCTS would likely require reasoning not just about the utility of each action, but also about the variance, since if we want to be reasonably sure that action a is better than a', we need to know that the lower bound of a is better than the upper bound of a'. However, I think this paper can serve as a good start to addressing this paper in the full MCTS setting.

Correctness: As far as I can tell

Clarity: Yes

Relation to Prior Work: Yes

Reproducibility: Yes

Additional Feedback: Typos: * sampled-based, sample-bases, ... -> sample-based


Review 2

Summary and Contributions: This paper presents an approach for explicitly generating candidate action sets for sample-based planning for domains with very action spaces. They claim the novelty of their approach is the idea of optimizing marginal value of newly sampled action and encouraging diversity of action selection rather than greedily maximizing value or information gain.

Strengths: The idea of using marginal utility is not particularly complex, and it would be surprising if this had not been done in some way previously. But it is a reasonable approach to try. The paper claims to show what is an important result, that action generation could be trained while optimizing for marginal utility would lead to superior planning policies.

Weaknesses: Their two simple simulators which they use is explained in some detail, but the experimental setup could be described in more detail, particularly their parameter sweep. They also don't adequately explain many of their other design choices such as only taking 4 samples from the policy during learning and yet training 20,000 iterations for each state. The simplified curling simulator is reduced to the final choice of the last agent, this seems to be that it essentially becomes a single step game does it not? It is also referred to in the conclusion as "the Olympic game of curling" which is quite an exaggeration given the simplicity of the simulation.

Correctness: In the introduction, they make the slightly strange statement that continuous action spaces are hard to optimize because exhaustive search is "intractable". There is a difference between intractable and impossible and this sentence only confuses the motivation for their work.

Clarity: Their paper is fairly well laid out, with a good progression of methods explained in a useful way for the reader. At the end of section 6.1 they refer to KR-UCT when I believe they mean KR-UCB.

Relation to Prior Work: In the introduction, they say "The key insight in this work is that sampling from a good policy does not necessarily produce a good set of actions for planning", this seems to require further explanation for the reader as it is quite a broad claim and essential to their paper's motivation. They claim that their approach of a marginal utility objective is analogous to how cross-entropy loss is more stable than 0-1 loss. I don't know if this is the case, their experiments do seem to demonstrate this but they are on relatively small domains.

Reproducibility: No

Additional Feedback: After author rebuttal: The authors's rebuttal was clear and helpful. Applying this approach to an MCTS algorithm would be a great thing to see. Also, their comments on increasing the diversity of action selection to encourage exploration are encouraging.


Review 3

Summary and Contributions: Post-rebuttal: Thank you for detailing the main issues I raised in your rebuttal. Both responses are insightful and deserving of inclusion in the paper/appendix. ---------------------- The paper presents a new method for computing a candidate set of actions when the action space is too large or continuous. By focusing on selecting actions that maximize the unique utility (compared to the set of actions selected already). The introduced optimization term is motivated by the ultimate aim to return the maximum action in the set of candidates, while at the same time providing a means for gradient signal to be used across all (or many) of the actions in the candidate set.

Strengths: The insight used for this work is simple and effective. Arguing against the use of sum, and generalizing from max to mu, was a compelling way of presenting the work. Another key strength is that both discrete and continuous action spaces are considered. Had one been left out, I think it would indicate a major weakness. Finally, on the domains considered, the approach performs particularly well. In particular, for a score of ~1.5 in curling, it takes about 5x the number of actions in the candidate set for the policy-based approach (which is still widely adopted) to match the performance of mu.

Weaknesses: Perhaps one of the largest weaknesses of the work is the lack of exposition on the mismatch between the general rhetoric of an action set (being just that -- a set) and the approach that is proposed. The mu objective has every new action dependent on the previous ones, and so the order of actions makes a large difference. In the extreme, if the actions are sorted based on increasing utility, then gradient will be passed through each and every one, while the reversed list will see only the best action receive any signal (and thus be equivalent to the max utility). This conflation of a set viewpoint with the actual list realization needs to be explored and explained. Perhaps the authors have already done this, but it is not reflected well in the paper. I see two further weaknesses with the work as it stands: limited analysis of the domains and uncertainty in how this would behave when used in conjunction with a larger system. Re: Domains As mentioned above, it is indeed a strength that both discrete and continuous action spaces are considered. And while I'm well aware that its far too easy for reviewers to just request "do more experiments", I think it really would support the claims here. The two evaluations are quite primitive in terms of the phenomenon that is being described, and at least one more complicated example for either setting (continuous or discrete action choices) would establish that this isn't a one-off improvement. Re: Integration I agree with the authors that the choice of action is orthogonal to the other improvements and techniques that go into a larger RL system. But what is entirely unclear is if the strength of what is being proposed is entirely mitigated when used in conjunction with another approach. Would the MCTS setting or more sophisticated ExIt type of strategy benefit from what's being proposed here? I hope so, but really can't say given what is presented.

Correctness: The theory, claims, and empircal methods all seem correct as far as I can tell.

Clarity: I found the paper to be very well written, and the concepts explained in clear detail. My only (minor) recommendation would be to place the finer experimental details (architecture used, hyperparameters, etc) in a table in the appendix. It doesn't really add much to the main content of the paper.

Relation to Prior Work: I think a comparison using a larger more established system for multi-step reasoning should have been done. Alpha-* / ExIt style setup, or similar, would have gone a long way to show how this work can benefit the leading approaches in the field.

Reproducibility: Yes

Additional Feedback: There are two key topics that I would like to suggest the authors respond to both during the rebuttal (space permitting) and in the paper: ordering and diversity. Ordering: As mentioned as part of the weaknesses, the ordering is something unique for mu when considered in relation to the max or sum objectives. Is the approach learning to order the actions? If you were to lay out all of the actions sorted based on their Q value, where do those selected come from? Max and sum would push it to the extreme end, but it's unclear if mu would do that as well. A deeper analysis on the actual set (or list) of actions chosen (in relation to their Q value compared to those actions not chosen) could shine quite a bit of light on the understanding of what is happening here. Also, given the ordered nature of the action subset, using recurrent model seems like a natural candidate for this -- iteratively selecting the next action based on those already selected. Diversity: I think this is a core insight to the work, and the details provided in the appendix should be brought into the main paper if at all possible. The question remains, however, if diversity is truly what is providing the increased performance. What about selecting the actions to be maximally diverse rather than based on Q values? Recent works in planning have found that focusing on the novelty of the state can be extremely effective in avoiding local minima for the search to a solution (heuristic-free techniques can succeed in achieving the goal just by searching for states that are novel). If you can establish that diversity is indeed something that greatly improves the learning process -- even if you need to "cheat" in order to compute the diversity in a restricted setting -- and then further establish that the mu objective is reflecting this diversity, then it would strengthen the paper a great deal.

[Author Response · NeurIPS 2020]

We would like to thank the reviewers for their insightful comments.

Addressing the common point of limiting our experimentation to a single-decision setting, our intent was to focus our analysis only on the effects of candidate generation. By removing the influences of other factors on the performance of search, for instance, rollout policies and state value function approximations, we can focus the evaluation. We are aware that the sequential-decision setting requires extra reasoning. We would argue, though, that the other components of learning algorithms for search try to ameliorate the amount of reasoning needed --- indeed, learning a perfect value function approximation would essentially reduce a sequential-decision problem to a single-decision problem. However, we do plan on examining our ideas in a full MCTS setting, which we think is a problem deserving its own investigation.

With regards to the ordering of actions produced by marginal utility, it is indeed true that the gradients use the ordering that comes from the action index. However, the actions are not pre-sorted before the marginal utility gradient is calculated. Rather, by optimizing the marginal utility objective the model learns an inherent ordering of the actions. Even though our search ignores this ordering at present, we see potential in its use for progressive widening or alpha-beta pruning in sequential search. Plotting out the frequency with which the actions are selected (see below) over test states in the curling domain shows that in expectation, the first action generated is most often selected (although still under 40% of the time), then the second, and so forth. We will add this to the appendix of the final version of the paper.

We had run preliminary experiments in the curling domain that explicitly tried to optimize for diversity in the action set (as a penalty on the set similarity added to the sum objective) before we even explored the marginal utility objective. Simply put, the results were unimpressive, performing no better than without the diversity term in the objective. Furthermore, we found the action set to be very sensitive to the coefficient on the penalty. Too low and actions would just cluster around the single best action; too high and actions would spread to the corners of the action space. We believe that while a lack of diversity makes for an ineffective candidate generator for search, diversity alone is also not the answer (note that the max objective in our experiments does produce diverse but still ineffective candidate sets). Marginal utility seems to produce a set of complementary actions and not just different ones (i.e., covering situations that the previous actions do not account for). We can include the diversity-modified objective as another baseline in the final paper to make this more clear.

[Meta-Review · NeurIPS 2020]

The reviewers all agreed that the paper is making a solid contribution. The simplicity of the approach in this case was a positive, especially given the demonstrated effectiveness.